# Variability in Provider Assessment of Sepsis and Potential of Host Response Technology to Address this Dilemma—Results of an Online Delphi Study

**DOI:** 10.3390/jpm13121685

**Published:** 2023-12-05

**Authors:** Chadd K. Kraus, Hollis R. O’Neal, Nathan A. Ledeboer, Todd W. Rice, Wesley H. Self, Richard E. Rothman

**Affiliations:** 1Department of Emergency and Hospital Medicine, Lehigh Valley Health Network (LVHN), University of South Florida (USF) Morsani College of Medicine, Tampa, FL 33602, USA; 2Department of Critical Care Medicine, Louisiana State University, Baton Rouge, LA 70803, USA; honeal@lsuhsc.edu; 3Department of Pathology & Laboratory Medicine, Medical College of Wisconsin, Milwaukee, WI 53226, USA; 4Division of Allergy, Pulmonary, and Critical Care Medicine, Vanderbilt University Medical Center, Nashville, TN 37232, USA; todd.rice@vumc.org; 5Department of Emergency Medicine, Vanderbilt Institute for Clinical and Translational Research, Vanderbilt University Medical Center, Nashville, TN 37232, USA; 6Department of Emergency Medicine, Johns Hopkins University, Baltimore, MD 21287, USA

**Keywords:** sepsis, triage, emergency department, host response testing, sepsis protocol

## Abstract

Potentially septic patients have a huge clinical and economic impact on hospitals and often present to the emergency department (ED) with undifferentiated symptoms. The triage of these patients is complex and has historically relied heavily upon provider judgment. This study aims to evaluate the consistency of provider judgment and the potential of a new host response sepsis test to aid in the triage process. A modified Delphi study involving 26 participants from multiple specialties was conducted to evaluate provider agreement about sepsis risk and to test proposed actions based on the results of a sepsis test. The participants considered case vignettes of potentially septic patients designed to represent diagnostic dilemmas. Provider assessment of sepsis risk in these cases ranged from 10% to 90% and agreement was poor. Agreement about clinical actions to take in response to testing improved when participants considered their own hypothetical borderline cases. New host response testing for sepsis may have the potential to improve sepsis diagnosis and care and should be applied in a protocolized fashion to ensure consistency of results.

## 1. Introduction

Sepsis is a life-threatening medical emergency that requires urgent intervention to achieve optimal clinical outcomes [1]. Despite its high prevalence, morbidity, and morality, sepsis is commonly misdiagnosed or has delayed diagnosis resulting in both over and under-treatment of sepsis [2,3], due to both a clinical presentation that is often nonspecific and lack of a definitive diagnostic test [4,5,6]. The combination of difficulty in initial diagnosis, coupled with the high prevalence of this condition, and need for urgent interventions to improve outcomes makes sepsis one of the most costly conditions facing U.S. hospitals [7].

A complicating factor in sepsis diagnosis and management is that the definition of the disease has been in evolution over the past three decades. The current definition is life-threatening organ dysfunction caused by a dysregulated host response to infection [8]. Although this updated definition is helpful with regard to understanding the consequence of the condition, making a clinical diagnosis of sepsis at the bedside would ideally be possible prior to overt life-threatening organ dysfunction.

Currently, there are no widely accepted or available tests to reliably quantify the host immune response nor a standardized approach to employ such tests, so the process of early diagnosis is largely dependent upon provider suspicion, intuition, and clinical judgment [2,9]. Notably, this can differ substantially between providers and results in significant variability in making timely diagnosis and improving clinical outcomes [10]. Additionally, over 80% of patients with sepsis initially present to the emergency department (ED) [11,12], which has a dynamic fast-paced environment where clinical teams are faced with high volumes of acutely ill and injured patients, a vast number of decision nodes, and limited resources with which to triage, treat, and assign disposition. Without a “gold standard” for sepsis diagnosis, ED providers are often tasked with making an empiric diagnosis and initiating treatment with incomplete evidence using only initial presentation and history as inputs [2].

Because sepsis is variable in its presentation and no single, definitive diagnostic test exists, opportunities remain to improve the consistency and timeliness of diagnosis, particularly for patients who present to the ED. One group of diagnostic tests that may provide support for the challenges associated with timely and accurate diagnosis of sepsis is rapid host response diagnostic assays [13], that provide information about whether there exist indicators of a dysregulated host response (often prior to organ dysfunction); such tests could be used to trigger immediate, early, aggressive treatment decisions, and potentially improve patient outcomes. We previously reported aggregate findings from a modified Delphi study with an expert panel describing the needs for and potential uses of a rapid sepsis diagnostic in the ED [14].

In this manuscript, we aim to explore two specific case presentations to (1) fully characterize and evaluate the extent and nature of variability in expert clinician assessments or pre test probability of sepsis prior to the availability of a sepsis diagnostic and (2) discuss impressions from the expert panel regarding potential ways a specific rapid host response test (IntelliSep, Cytovale Inc., San Francisco, CA, USA) might be used to increase clinician consistency in the approach to sepsis diagnosis and subsequent clinical care.

## 2. Materials and Methods

### 2.1. Overview of IntelliSep Delphi Study

We previously reported findings from a modified Delphi study fielded online and undertaken with the explicit goal of reaching consensus on defining: (1) key unmet needs in sepsis diagnostics; and (2) the potential utility for a future hypothetical rapid host response sepsis test for use in the ED setting. Briefly, this study involved two sequential questionnaires that were completed by an expert consensus panel of 26 participants from disciplines involved in the care of sepsis patients, including Emergency Medicine (62%), Critical Care Medicine (19%), and Laboratory Medicine (19%). The specific process and thresholds for agreement are described in detail in the previous publication [see Appendix A] [14].

### 2.2. Description of IntelliSep Index 

The rapid host response test that the group chose to center the research around was Cytovale’s IntelliSep test, which received FDA clearance in December 2022. IntelliSep is a semi-quantitative test that assesses cellular host response via deformability cytometry of leukocyte biophysical properties [15,16]. The test is intended for use, in conjunction with clinical assessments and laboratory findings, to aid in the early detection of sepsis with organ dysfunction in adult patients presenting to the emergency department (ED) with signs and symptoms of infection. The test is performed on an EDTA anticoagulated whole blood sample and returns results in under 10 min.

The IntelliSep test generates an IntelliSep Index value, ranging between 0.1 and 10.0, that is divided into three discrete interpretation bands based on the probability of sepsis with organ dysfunction manifesting within the first three days after testing (Table 1). Band 1 represents a score range of 0.1–4.9, indicating a low (<5%) probability that the patient has or will develop sepsis with organ dysfunction. Band 2 represents a score range of 5.0–6.2, indicating a moderate (18–25%) probability that the patient has or will develop sepsis with organ dysfunction. Band 3 represents a score range of 6.3–10.0, indicating a high (44–48%) probability that the patient has or will develop sepsis with organ dysfunction. The naming convention of the IntelliSep Interpretation Bands was updated from Green, Yellow, and Red to Band 1, Band 2, and Band 3 in December 2022.

### 2.3. Patient Case Vignettes

Two hypothetical case vignettes (Patient A and Patient B) describing patients with suspected or documented infection that would present a “diagnostic dilemma” in the ED were included in both questionnaires (Table 2). These case vignettes were developed by the Delphi Steering Committee and were intended to reflect clinical presentations consistent with patients who may be considered as having an intermediate probability for sepsis in which diagnostic uncertainty or variability in clinical decision-making and practice may exist. Panelists were also asked to provide an example of a clinical scenario (Patient C) for a “borderline patient” (i.e., clinical uncertainty of sepsis diagnosis) where they would utilize the IntelliSep test.

### 2.4. Panelist Estimates of Sepsis Risk 

Panelists were asked to specify a pretest probability of sepsis for Patient A and Patient B having sepsis. The question was open-ended to allow the panelists to specify any pretest probability between 0% and 100%. Panelists were also able to provide commentary around their clinical impressions of both Patients A and B (Table 3). In addition, panelists ranked their level of agreement using a 5-point Likert scale (e.g., Strongly Disagree, Disagree, Neutral, Agree, Strongly Agree) with Delphi statements associated with the IntelliSep Index for each Patient case.

### 2.5. Data Analysis 

Data for pretest probability estimates for Patients A and B were aggregated for participants of differing medical specialties. Pretest probability reported by the panelists was compared across different medical specialties with statistical significance between groups in calculated using analysis of variance (ANOVA) with a Tukey HSD post hoc test. For Patient C, where participants were asked to describe a “borderline patient” (i.e., clinical uncertainty of sepsis diagnosis), standardized keywords were assigned to the descriptions that participants provided and were counted to analyze trends in patient description. 

## 3. Results

### 3.1. Variability among Panelists for the Diagnostic Scenarios Posed by Patient A and Patient B

We observed a wide range (10–100%) in perceived pretest probabilities for both Patient A (Figure 1) and Patient B (Figure 2). For Patient A, 38% of panelists specified a pretest probability of sepsis of less than 50%, 12% of panelists specified a 50% pretest probability, and 50% of panelists specified pretest probability of more than 50%. For Patient B, 31% of panelists specified a pretest probability of sepsis of less than 50%, 27% of panelists specified a 50% pretest probability, and 42% of panelists specified pretest probability of more than 50%. No trend in pretest probabilities across specialties was observed for either Patient A or Patient B (Appendix A).

### 3.2. Influence of Initial Clinical Impression on Consensus Recommended Clinical Actions with Incorporation of the IntelliSep Test

There was substantial variability observed among panelists’ clinical impressions of both Patient A (Table 4) and Patient B (Table 5) as reflected in the written commentaries. While some panelists perceived a high risk of sepsis, others considered the patients as not having sepsis or with an unclear diagnosis of sepsis. The expert panelists also differed in their views about the IntelliSep Index for Patient A and Patient B (Appendix A). Notably, the panelists expressed views that were consistent with their clinical impression of each patient. For example, some panelists found a Band 1 result reassuring if they thought the patient’s risk of sepsis was low, while other panelists found a Band 3 result helpful in supporting their view that the patient was at high risk for sepsis. The panelists also differed in how they viewed a Band 2 result. Some panelists who perceived a low risk of sepsis thought a Band 2 result may support a decision to observe the patient more closely, while panelists who perceived a high risk for sepsis thought a Band 2 result would not alter their clinical management of the patient. A few panelists expressed the view that they would not change clinical practice regardless of the test result, either because they were inclined to pursue care based on their perceived risk of sepsis or, in one case, because they generally do not have access to or utilize specific pathogen testing.

### 3.3. Consistency among Panelists for Self-Generated “Borderline” Patient C

Subsequent to their evaluation of Patients A and B, panelists were asked to provide an example of a clinical scenario for a “borderline patient” (i.e., clinical uncertainty of sepsis diagnosis) to isolate the variability inherent in the perceived risk of sepsis and evaluate the potential of a host response test like IntelliSep to drive consistency of action.

### 3.4. Commonalities in Borderline Patient C Descriptions

Each expert panelist generated their own description of Patient C that they considered to be borderline for sepsis. Although these descriptions were created independently and not shared with other panelists, there were some common clinical themes and characteristics (Figure 3). For example, many of the expert panelists described demographic information such as the age and sex of their borderline Patient C. Of the 15 panelists that considered age, 8 described elderly patients (i.e., 65 years of age or older). Of the 8 panelists that specified the sex of their borderline Patient C, 7 described females and one described a male. Most (16 of 26) panelists included one or more systemic inflammatory response syndrome (SIRS) criterion in their borderline Patient C descriptions, including temperature (10 elevated, 1 hypothermia), heart rate (8), respiratory symptoms (7), and white blood cell (WBC) count (6). All 26 panelists included additional test results and observations including blood pressure (*n* = 10), comorbidities (8), uncertain etiology (5), mental states (4), lactic acid (4), pneumonia (2), COPD (2), UTI (2), and renal disease (2).

Expert panelists were asked to rank their level of agreement with Delphi statements about IntelliSep Index results using the context of their borderline Patient C. Unlike what was observed for Patient A and Patient B, consensus was relatively high among panelists for the characteristics for their borderline Patient C. Consensus was reached for three (60%) of the five statements where the IntelliSep Index was “Band 1”, all four (100%) of the four statements where the IntelliSep Index was “Band 3”, and two (40%) of the five statements where the IntelliSep Index was “Band 2” (Appendix A). Among the nine Delphi statements where consensus was reached, panelists agreed with eight of the statements and did not agree with one of the statements (Table 6).

The relative consistency in how the panelists ranked the Delphi statements for their borderline Patient C was also apparent in the written commentaries (Table 6). In contrast to responses for Patients A and B, all 26 panelists found that a Band 1 result would reassure them that their borderline Patient C did not have sepsis and would give them more clinical confidence in pursuing alternate diagnosis and treatment. All but one of the panelists found that a Band 3 result would encourage them to take a more aggressive approach in treating their borderline Patient C for sepsis. The remaining panelist did not consider a Band 3 result to be helpful, since they were already thinking that their borderline patient had a 50% probability of sepsis. The expert panelists differed in how they viewed a Band 2 result. Some of the panelists thought that a Band 2 result was less helpful in diagnosing whether or not their borderline Patient C had sepsis, while other panelists thought a Band 2 result would prompt them to take a more aggressive approach and/or to repeat IntelliSep to see whether there is any change in the test result over time.

## 4. Discussion

This study provides insights into the challenges related to the diagnosis and management of sepsis. We found baseline significant variability among providers in assessing the pretest probability for sepsis in two hypothetical patients with common clinical scenarios. The determination of pretest probability, or the estimated probability that a patient has the disease in question before a diagnostic test is performed, is a crucial aspect of the diagnostic process because it impacts the interpretation of and, thus, action on the result of any given test [17]. Thus, variability in the determination of pretest probability results in variability in both diagnostic test utilization and interpretation and ultimately in treatment decisions. Our findings are similar to those of Rhee et al. [18], who noted significant variability in clinicians’ diagnosis of sepsis. Unlike Rhee et al., who used entire records in their case vignettes, we used only brief vignettes with basic laboratory data to mimic what would be available to ED providers early in the course of the ED visit. These findings suggest that provider gestalt is inconsistent as a means for diagnosing sepsis and highlight the necessity of developing a more standardized process for both the implementation of a diagnostic aid and the initiation of treatment for sepsis.

In a previous published study using the same set of panelists, we asked the experts to provide their feedback on the ideal properties and potential use cases for a sepsis diagnostic test [14]. Participants indicated they would like a test with a 95% negative predictive value in a population of patients with an estimated pretest probability between 25% and 68%. Notably, there was significant variability in what pretest probability of sepsis each panelist thought was appropriate for using the test. There was agreement, however, that that an ideal sepsis diagnostic would best be deployed early in an ED course and have rapid turnaround time (<30 min).

Variability in both the assessment of pretest probability, as well as variability of what threshold would be appropriate for a provider to order such a test, may complicate the implementation of an adjunctive sepsis diagnostic test, making the systemic benefit of the test more difficult to achieve. For example, in the provided cases, the average pretest probability for Patient A was 53.7%, and for Patient B it was 54.0%, indicating that the collective group of experts felt that these cases represent ideal borderline patients for the application of a sepsis diagnostic. Further evaluation, though, reveals considerable variability among providers in this estimate that could impact application of the test in practice. In this study, 15 of 26 providers perceived a pretest probability for Patient A that fell within the pre-specified pretest probability range panelists identified for applying such a test (25–68% probability of sepsis). This indicates that only 15 panelists would have used the test for Patient A and would, presumably, allow the result to guide therapy. Similarly, for Patient B, only 14 of 26 would have fallen within the same pretest probability range. This suggests a potential for inconsistent application of the test that would hamper efforts in having the test consistently used to drive more consistent clinical care. Ironically, panelists in the Delphi study did not perceive that variability of sepsis care was a significant issue. That said, a majority of respondents indicated that improving sepsis outcomes and resource utilization are important downstream effects of a sepsis diagnostic. When taken in total, our study suggests that reliance on provider intuition alone for the early assessment of sepsis and use of sepsis diagnostics may contribute to misdiagnosis. This argues for a more systematic application of such testing, which may allow centers to achieve resource and outcome targets despite variability in provider experiences and perceptions.

Interestingly, when the expert panel themselves developed a self-defined a hypothetical borderline sepsis patient (Patient C), the amount of variability in pretest probability assessment decreased substantially as evidenced by the amount of consensus reached with regards to clinical actions based on the IntelliSep result. This exercise shows that clinicians are more likely to act consistently on a diagnostic test when they perceive a diagnostic dilemma, despite the breadth of characteristics in Patient C examples. As an example, though many respondents described the age of Patient C as a factor, they were evenly divided between those who chose ≥ 65 (53%) and those who chose <65 (47%). Also, SIRS symptoms were frequently mentioned (most commonly temperature) but potential infectious etiology and other characteristics varied widely. These results indicate that a “borderline” patient can assume a variety of presentations when seen from the perspectives of different providers—findings consistent with the heterogeneous and nonspecific presentation of sepsis. The variability in pretest probability of sepsis observed in this study suggests concomitant heterogeneity in providers’ intuition, frequently resulting in failure to consider sepsis as an option or, conversely, to focus too narrowly on sepsis as a diagnosis. This work confirms that there do exist biases in provider perceptions of sepsis at all levels, influenced by training and experience, that do impact clinical decision-making. In order for a sepsis test to truly impact care, there needs to be institutional protocolization of such a diagnostic so that the reliance on provider judgment and perception of sepsis, which this and past research has noted to be quite variable, is reduced.

To assist clinicians in the ED, consensus guidelines recommend formal sepsis programs for screening for high-risk patients and standardization of treatment; however, these guidelines do not provide guidance for implementation of such a program [19]. One potential future solution for improving sepsis diagnosis is to implement a standardized process for use of a diagnostic test in the population of patients who present to the ED with suspicion for infection. A standardized process allows for continuous assessment of test performance and subsequent clinical action and outcomes [20]. In this context, the process could be adjusted to identify the appropriate population of patients for application of the diagnostic test, resulting in desired conservation of time and resources [21] and optimization of outcomes. Furthermore, because the clinical course of patients and clinical data evolve throughout an ED encounter, the process should allow for continuous assessment of the patient and application of the diagnostic test rather than a point assessment at one time during an ED visit (for example, triage). Finally, a standardized process for implementation of a sepsis diagnostic may allow for analysis of prevalence within this population and may facilitate the adoption of care pathways based on diagnostic test results similar to structured processes for the evaluation and management of patients presenting to the ED with chest pain.

Considering Patients A and B from our study can provide context for the implementation of such a process. Both patients had elements of SIRS symptoms with the potential for infection. Previous studies have shown that evaluating patients presenting to the ED with two or more SIRS criteria captures approximately 85% of patients presenting to the ED with sepsis [22]. A triage process that includes the vital sign components of the SIRS criteria, plus an initial clinical pretest assessment for suspicion of infection, may provide a first line of detection. While “suspicion of infection” may introduce a level variability similar to that documented in this study, the broader nature of infection versus sepsis may allow for inclusion of a sufficient selected group of potential sepsis patients. As the ED visit evolves, other criteria for including a sepsis diagnostic will provide further opportunity for detection of sepsis. These criteria might be another provider’s suspicion or laboratory data related to infection (i.e., complete blood count and/or differential results). Other screening tools, including scoring systems [23,24,25,26], artificial intelligence [27], and electronic health record based systems [28] may be included as well. Once applied to a standardized population, the results of a diagnostic could be more easily applied in protocolized care and/or care pathways. Integration of a sepsis diagnostic such as IntelliSep into clinical management could potentially impact a variety of aspects in sepsis care including patient outcomes, resource utilization, and antimicrobial stewardship [29]. Indeed, a recent costs and consequences modeling study found that the IntelliSep test has the potential to both lower mortality and lower costs of care on average relative to a strategy employing Procalcitonin [30]. The IntelliSep test, as performed on a whole blood sample, with a blood to answer timeframe of under ten minutes, may allow for a more seamless integration into an ED protocolized workflow and augment provider decision-making in real-time.

### Study Limitations

Our study has limitations. Although we surveyed a variety of experts, all considered to be stakeholders in a hospital-wide sepsis program, the total number of providers included is relatively small, prohibiting statistical assessment or comparison of the different disciplines. Also, we included only two case studies, both of which illustrate common presentations, so we are unable to assess participants’ responses to a wide variety of cases and presentations as would typically be seen in the ED setting. In addition, this study was designed to gather input on recommended use of a specific sepsis diagnostic test (IntelliSep) by the sponsor of this study (Cytovale Inc.) and does not consider a broad range of potential tests. Finally, participants were aware that the topic of discussion was to explore the optimal characteristics and utility of a sepsis diagnostic, and this prior knowledge may have influenced their decisions and assessment of pretest probability.

## 5. Conclusions

A rapid, reliable diagnostic aid is needed to assist providers in the efficient and effective delivery of sepsis care. Given limited case vignettes of common presentations that include data available early in the course of an ED visit, provider assessment of pretest probability varies widely, which can have implications for the diagnosis of sepsis and subsequent timely and appropriate patient care. We believe these findings highlight the need for improved diagnostic tools such as host response assays in the ED setting to aid in the rapid diagnosis and risk stratification of patients suspected of sepsis, along with the need for these tools to be implemented in a protocolized fashion to address the variability among providers in identifying sepsis.

## Figures and Tables

**Figure 1 jpm-13-01685-f001:**
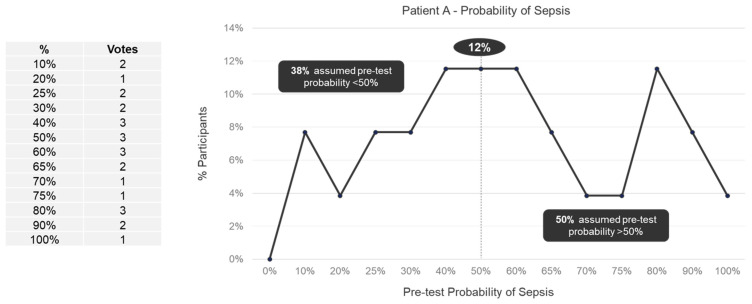
Pretest probability of sepsis for Patient A among expert panelists (*n* = 26).

**Figure 2 jpm-13-01685-f002:**
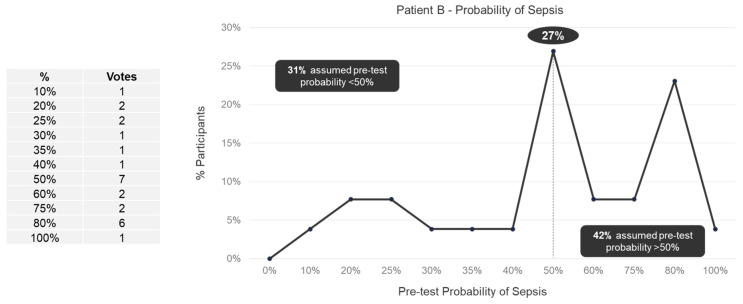
Pretest probability of sepsis for Patient B among expert panelists (*n* = 26).

**Figure 3 jpm-13-01685-f003:**
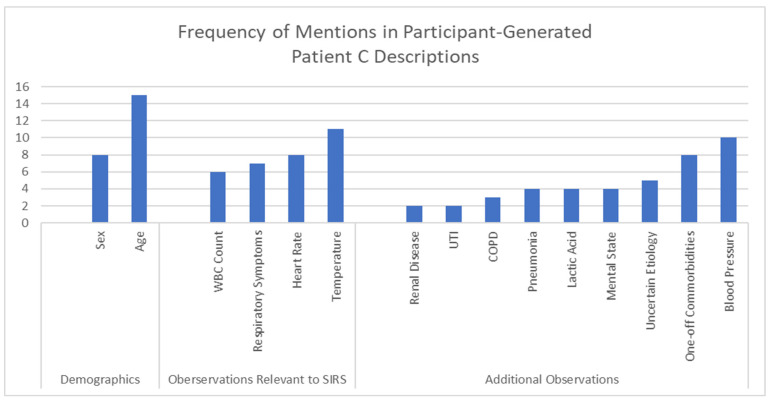
Common elements in self-generated descriptions for borderline Patient C.

**Table 1 jpm-13-01685-t001:** IntelliSep Index Value Bands.

IntelliSep Test Interpretation Band	ISI Range	Results Interpretation Considerations
BAND 1 (Low Probability of Sepsis)	0.1–4.9	All results should be interpreted in the context of the other clinical observations and laboratory test results for the patient.
BAND 2	5.0–6.2	
BAND 3 (High Probability of Sepsis)	6.3–10.0	

**Table 2 jpm-13-01685-t002:** Case vignettes for patients presenting a “Diagnostic Dilemma” for sepsis.

Case	Demographic Information and Patient History	Vital Signs	Additional Observations
Patient A	72 year-old female Nursing home patient History of dementia, hypertension, and dyslipidemia	Temperature 97.8 °F Pulse 84 bpm Respiratory rate 16 breaths per minute Oxygen saturation 95% on room air Blood pressure 98/62 mmHg WBC 9.8 K/μL	Altered mentation BUN 32 mg/dL Creatinine 1.9 mg/dL (baseline 0.8 mg/dL) Lactate 2.8 mmol/L Urinalysis of a catheter specimen revealed nitrite-positive urine, 6–10 WBC/HPF, 0–5 RBC/HPF, and many bacteria on microscopic exam
Patient B	65 year-old male History of ischemic cardiomyopathy (ejection fraction 30% on echocardiogram 6 months prior), cocaine abuse, chronic kidney disease, hypertension, poorly controlled diabetes mellitus, and multiple hospitalizations for heart failure exacerbation associated with cocaine use	Temperature 100.2 °F Pulse 98 bpm Respiratory rate 22 breaths per minute Oxygen saturation 92% on 4 L/min oxygen delivered by nasal cannula Blood pressure 148/92 mmHg WBC 12.6 K/μL (12% bands)	BNP 2360 pg/mL BUN 46 mg/dL Creatinine 2.3 mg/dL (baseline 1.5 mg/dL) Lactic acid 3.8 mmol/L Rales in the right base on physical exam, as well as 2+ bilateral lower-extremity edema X-ray reveals an enlarged cardiac silhouette, vascular congestion similar to previous exams and a new, hazy right lower lobe alveolar infiltrate Shortness of breath and chest pain associated with a nonproductive cough for the past 2–3 days

**Table 3 jpm-13-01685-t003:** Clinical impressions provided for Patients A and B from panelists.

Topic	Selected Quotes from Expert Panelists
Clinical impression—Patient A	“*I think Patient A is likely not septic but has dehydration possibly related to a urinary tract infection.*”
	“*This patient is at higher risk (pretest probability based on demographics and clinical gestalt) for infection as a cause of presenting symptoms.*”
	“*To me, this patient is a tweener. Older patient, so at risk, with lowish BP in the 90s’ but otherwise stable temp and pulse, altered mental status, and high lactate with a possible urinary source.*”
Clinical impression—Patient B	“*Patient is sick and has a high chance of sepsis.*”
	“*Would likely still do CXR to look for focal infiltrate given exam -but in general, less worried about sepsis.*”
	“*Likely pneumonia superimposed on CHF, not clearly sepsis.*”

**Table 4 jpm-13-01685-t004:** Representative written commentaries from expert panelists about Patient A.

Topic	Quotes from Expert Panelists
Value of IntelliSep Index “Band 1”	“*I think a [Band 1] would reinforce my hypothesis that this patient is dehydrated, possibly related to a urinary tract infection.*”
	“*While a [Band 1] result might make me feel a bit better about her prognosis, given the high pretest probability I would probably still treat her as septic until proven otherwise.*”
Value of IntelliSep Index “Band 3”	“*Regardless of the results, there is simply not enough capacity for patients like this in our healthcare system, an ICU or step-down unit. At best, more frequent vital sign assessments or neurological check might be reasonable. In general, I do not have access to nor utilize aggressive pathogen ID testing, and it does not influence my decision making.*”
	“*The [Band 3] test pushes my suspicion of sepsis much higher and could supplant longer observation and repeat lactate, etc.*”
Value of IntelliSep Index “Band 2”	“*Since I think patient A likely has a urinary tract infection and dehydration, a [Band 2] may convince me to be a bit more aggressive in the context of observing her more closely if other data support doing so.*”
	“*I think in this patient with a high pretest probability of sepsis, a [Band 2] would not change my management very much.*”
	“*I would not consider sending this patient home regardless of [Band Number].*”

**Table 5 jpm-13-01685-t005:** Representative written commentaries from expert panelists about Patient B.

Topic	Quotes from Expert Panelists
Value of IntelliSep Index “Band 1”	“*This is the exact patient a [Band 1] would be the most helpful, one with CHF and real concern for fluid resuscitation consistent with sepsis. My pretest probability is higher for alternative diagnosis as well.*”
	“*In this complicated patient who meets several criteria, I do not think the [Band 1] would be particularly helpful in decision making.*”
Value of IntelliSep Index “Band 3”	“*[Band 3] in this patient would be helpful to assist in distinguishing heart failure exacerbation alone versus sepsis associated symptoms.*”
	“*I would treat this patient for presumed sepsis no matter what, but a [Band 3] would reaffirm this and probably give me even more urgency (as it’s possible to be slightly reassured by his normal blood pressure on presentation).*”
	“ *[Band 3] would suggest this patient is infected -but still worried about volume overload and need to adjust clinical care (i.e., resuscitation, etc.) with that in mind.*”
Value of IntelliSep Index “Band 2”	“*A [Band 2] may indicate that this patient is on the sepsis trajectory and may be helpful in supporting an aggressive approach when used with other clinical information.*”
	“*[Band 2] is not clinically helpful in this case.*”
	“*I would not consider discharging this patient regardless of the test result.*”

**Table 6 jpm-13-01685-t006:** Recommended actions based on IntelliSep results for a borderline patient (Patient C).

Band 1	Band 2	Band 3
Panel Agreement with the following statements: Give additional consideration to an alternate diagnosis (other than sepsis) Weigh the risks of anti-microbial stewardship in the determination of antibiotic protocol (e.g., Could influence my decision to select a shorter duration, narrow spectrum, oral vs. IV) Consider more conservative fluid resuscitation	Panel Agreement with the following statements Continuing sepsis-related workup (watchful waiting) Tracking the trajectory of the IntelliSep scores over time (if data were available supporting the utility of doing so)	Panel Agreement with the following statements Immediate aggressive sepsis-focused care Immediate initiation of antibiotics Ordering blood cultures
		Panel Disagreement with the following statements: A high likelihood of an alternate diagnosis (other than sepsis)
Panelist comments related to a Band 1 result from the IntelliSep test for patient C: *“This is my ideal scenario, lower risk, healthier cohort with sepsis syndrome that I can limit work up and send home safely.”* *“No clear infection to suggest sepsis, negative test gives more confidence in impression.”* *“[Band 1] result would increase my concern for a non-infectious cause such as dehydration, medications, etc. Would pursue alternatives and withhold broad spectrum IV antibiotics.”* *“The test result would drive the practice—if [Band 1], supportive care for non-infectious aspiration pneumonitis.”*	Panelist comments related to a Band 2 result from the IntelliSep test for patient C: *“A [Band 2] in this case, while not as worrisome as red [Band 3], would still on balance be concerning, as a >20% probability of sepsis cannot be taken lightly. So it would probably push me to treat her on the aggressive side.”* *“A [Band 2] emphasizes to me that this borderline patient may be clinically evolving. I would start sepsis workup and perhaps repeat the IntelliSep test while observing closely and as more data becomes available.”*	Panelist comments related to a Band 3 result from the IntelliSep test for patient C: *“In this patient, IntelliSep would be confirmatory (i.e., rule-in) rather than exclusionary (i.e., helping to rule-out) sepsis.”* *“In this borderline patient with a high probability score I would take an aggressive approach.”* *“Would raise concern for infection and prompt typical ED sepsis management.”*

## Data Availability

Data are contained within the article and Appendix A.

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
