# Peer review of "Variability in Provider Assessment of Sepsis and Potential of Host Response Technology to Address this Dilemma—Results of an Online Delphi Study"

_jpm, 2023, doi:10.3390/jpm13121685_

Round 1
Reviewer 1 Report
Comments and Suggestions for Authors
The paper is well-written and makes a significant effort to ensure readability and comprehension. The study's primary aim is to assess the utility of a "new host response sepsis test" and explore its potential implementation in clinical practice.
As critical care physicians, we have all encountered clinical scenarios where a diagnosis of sepsis is made based on clinical intuition. It's common for different clinicians to have varying pre-test probabilities of sepsis, and even the same clinician may reassess findings differently at a later time. This study confirms the variability we observe daily in sepsis diagnosis. Therefore, the need for an objective test to support clinical judgment becomes evident.
What's intriguing in this study is that only 50-60% of clinicians showed a willingness to use the "host response test," and their interpretations of the test results were highly variable. These findings raise important questions about whether this test can genuinely influence clinical practice, such as reducing antibiotic use or minimizing diagnostic and imaging studies for sepsis source evaluation. Based on the study results, it appears that this test may not significantly impact these critical clinical parameters. Consequently, the paper's conclusion suggesting that host response assays be considered for use in the emergency department (ED) to assist in rapid sepsis diagnosis and risk stratification, while also addressing provider variability, may not fully acknowledge the limitations in the test's clinical utility.
A more comprehensive discussion of the test's limitations and challenges in practical implementation in the final conclusions is necessary for a clearer understanding of its real-world applicability.
Reviewer 2 Report
Comments and Suggestions for Authors
Title and abstract:
I would suggest that the title of the research should include where and what type of study was conducted in order to make it more accurate and comprehensive for readers.
It would be helpful if you expand the abstract conclusion in order to convey the current findings and suggestions to the stakeholders.
Inconsistency between the aim stated in the abstract and the introduction section.
Research Methodology: The approach is straightforward. However, the authors do not specify the sample size and the variation in participant numbers across hospitals. It is crucial to clarify how participants were selected from different hospitals, providing background information on the study sites. Additionally, a comprehensive explanation of the research design is needed in the methodology section.
Results and Discussion: The findings are promising, but the discussion section requires substantial reworking.
Conclusion: The conclusion is unclear and does not meet acceptable standards.
On the one hand, the study requires significant additional work, given that although there is a literature review, the theoretical background and the relevant research results are not analyzed in sufficient depth. On the other hand, a significant revision is also necessary because there are no exact research questions or hypotheses in the light of which the results could be interpreted. In addition to these observations, the study could still be improved, but I do not consider it worthwhile, since the research is entirely descriptive and provides little explanation for the causes of the phenomena. The number of processed literatures is extremely small, and the length of the study is also below the average
Kindly use this article
Nurses' knowledge, attitudes, practice, and decision-making skills related to sepsis assessment and management.
Round 2
Reviewer 2 Report
Comments and Suggestions for Authors
none